# Absence of a pressure gap and atomistic mechanism of the oxidation of pure Co nanoparticles

Jaianth Vijayakumar [1], Tatiana M. Savchenko[1], David M. Bracher[1], Gunnar Lumbeeck[2], Armand Béché [2], Jo Verbeeck[2], Štefan Vajda [3], Frithjof Nolting [1], C.A.F. Vaz [1] ✉ & Armin Kleibert [1] ✉

Understanding chemical reactivity and magnetism of $3d$ transition metal nanoparticles is of fundamental interest for applications in fields ranging from spintronics to catalysis. Here, we present an atomistic picture of the early stage of the oxidation mechanism and its impact on the magnetism of Co nanoparticles. Our experiments reveal a two-step process characterized by (i) the initial formation of small CoO crystallites across the nanoparticle surface, until their coalescence leads to structural completion of the oxide shell passivating the metallic core; (ii) progressive conversion of the CoO shell to $Co_3O_4$ and void formation due to the nanoscale Kirkendall effect. The Co nanoparticles remain highly reactive toward oxygen during phase (i), demonstrating the absence of a pressure gap whereby a low reactivity at low pressures is postulated. Our results provide an important benchmark for the development of theoretical models for the chemical reactivity in catalysis and magnetism during metal oxidation at the nanoscale.

The chemistry and magnetism of nanoparticles of $3d$ transition metals and metal oxides are of fundamental interest for applications in fields ranging from spintronics, as elemental building blocks or as components in novel high performance magnets[1,2]; to catalysis, including for water hydrolysis, Fischer–Tropsch reactions, or for controlling reaction kinetics using local magnetic heating protocols[3–7]; and in biomedicine, for example, as environmental assays, for applications in tissue imaging, or in hyperthermia for cancer treatment[8,9]. Such interest is a direct consequence of their reduced size and of the modified properties that emerge at the nanoscale induced by quantum confinement and dominant surface effects that lead to enhanced electronic and magnetic properties and to high chemical reactivity. The reduction in size to the nanoscale also permits the stabilisation of crystal structures not stable in bulk, while the presence of varying oxidation states of metal cations leads to a wide range of nanoparticle systems, each with its unique properties. Among these systems, cobalt and cobalt oxide nanoparticles in particular have led to important discoveries, such as

the nanoscale Kirkendall effect, which underpins the synthesis process of hollow nanostructures for nano-reactors or for complex nano-composite systems[10–12], and the magnetic exchange-bias effect[13], which has been fundamental for the development of today's spintronics devices[2,14–17]. However, and despite the immense body of work, basic questions about the actual oxide shell formation in cobalt nanoparticles and its impact on the magnetism of oxide-shell-metallic-core nanoparticles remain open. For instance, most investigations on the oxidation of cobalt nanoparticles report the presence of rocksalt cobalt oxide (CoO) shells surrounding a remaining metallic cobalt core or CoO hollow spheres, while the formation of the anticipated, thermodynamically more stable spinel $Co_3O_4$ is less frequently found[9,10,18–26]. Moreover, in most experiments, high or even ambient pressures and elevated temperatures were required to achieve a substantial oxidation of the Co nanoparticles. These findings are in stark contrast to the high reactivity of thin Co films to molecular oxygen, where minute oxygen exposures results in the formation of CoO and

[1]Swiss Light Source, Paul Scherrer Institut, 5232 Villigen PSI, Switzerland. [2]EMAT, University of Antwerp, 2020 Antwerpen, Belgium. [3]Department of Nanocatalysis, J. Heyrovský Institute of Physical Chemistry v.v.i., Czech Academy of Sciences, Dolejškova 2155/3, 18223 Prague, Czech Republic. ✉e-mail: carlos.vaz@psi.ch; armin.kleibert@psi.ch

subsequently of $Co_3O_4$, even at cryogenic temperatures and under vacuum conditions[27–29].

Such differences in chemical reactivity are often assigned to the so-called pressure gap, which refers to the long-standing problem of reconciling low and high pressure reactivity in heterogeneous catalysis[30–32] and attributed to mass transfer limitations, surface coverage with adsorbed species, the onset of chemical reactions that are absent at low pressure, the formation of different products under low and high pressure, variations in the oxidation of the catalytic metal, or to dependence of the reactivity on the oxidation state under different working pressure conditions[33]. Of particular impact to the atomic-level understanding of the pressure gap could be the presence of contaminants at higher pressures which are absent under ultrahigh vacuum conditions[30]. Concerning the magnetic properties, the reported structural and chemical properties hardly explain the magnitude of the experimentally determined exchange bias effects as well as other magnetic phenomena such as superparamagnetic inclusions or disordered spins in the oxide shell[2]. Finally, an experimental confirmation for the uniform oxidation process that is assumed in the oxidation theory of Cabrera and Mott frequently discussed in the context of the oxidation of $3d$ transition metal nanoparticles is still missing[2,25,34–39].

These ambiguities are compounded by the experimental difficulty of investigating the structure, composition, and magnetic properties of $3d$ transition metal nanoparticles during oxidation under well-defined conditions. For instance, it is well known that both the reactivity to oxygen and the magnetic properties of nanoparticles depend on a large variety of factors including particle size, shape, structure, separation, mobility, nature of the substrate or embedding medium, and reaction conditions, which makes it difficult to disentangle these different aspects. Also, the detailed structural characterisation of nanoparticles during the different stages of oxidation is challenging, since typical investigations require air exposure of the samples for the transfer to suitable high-resolution electron microscopes, while the vacuum conditions in most of these instruments are not suitable for well-controlled in situ oxidation studies. Moreover, it is known that Co nanoparticles with very distinct magnetic properties in the pristine metallic state can coexist, irrespective of particle size, which makes the interpretation of their magnetic properties upon oxidation complicated, in particular when averages over large ensembles are considered[40].

In this work, we take advantage of the capabilities of in situ x-ray photoemission electron microscopy (XPEEM)[41] for probing the electronic, chemical, and magnetic state of individual nanoparticles in combination with ex situ high-angle annular dark-field scanning transmission electron microscopy (HAADF-STEM) measurements to yield the atomically resolved structure of nanoparticles conserved in different oxidation states by means of passivating carbon capping layers[40,42–48]. This approach enables us to exclude effects of particle size, shape, mobility, and interparticle magnetic and chemical interactions to determine the intrinsic oxidation process of metallic Co nanoparticles and its impact on the magnetic properties. Our results unambiguously demonstrate that pure Co nanoparticles supported on chemically inert substrates exhibit a high reactivity to molecular oxygen, comparable to that of thin Co films under ultrahigh vacuum conditions. We argue that the very different oxidability of Co nanoparticles reported in the literature is related to their purity, rather than due to the presence of a pressure gap between low and high pressure experiments. Our structural characterisation shows that the oxide shell does not grow continuously, as frequently assumed in the literature and premised in the metal oxidation theory of Cabrera and Mott[49], but instead develops through the simultaneous nucleation and growth of independent CoO crystallites randomly distributed over the nanoparticle surface. In fact, we find that the oxidation reaction in Co nanoparticles occurs in a two-step process with an initial rapid formation of CoO crystallites, until coalescence closes the oxide shell, and

a subsequent progressive conversion of CoO to $Co_3O_4$. At this stage, the nanoscale Kirkendall effect sets in, which leads to a physical and magnetic decoupling of the oxide shell to the remaining metallic core, possibly explaining the reduced exchange bias effect observed in this system as compared to the theoretical value.

## Results

### Structure and magnetism in the pristine metallic state

Pure metallic fcc Co nanoparticles with sizes ranging between 10 and 20 nm are grown from the gas phase in an arc cluster ion source and deposited in ultrahigh vacuum (UHV) on chemically inert substrates containing marker structures for sample navigation. After deposition, the nanoparticles are characterised in situ by means of XPEEM, Fig. 1a. The chemical state of the nanoparticles directly after deposition as well as after step-wise oxidation is characterised by means of local x-ray absorption (XA) spectroscopy using linearly polarised x-rays. The magnetic properties are probed using circularly polarised x-rays by taking advantage of the x-ray magnetic circular dichroism (XMCD) effect at the Co $L_3$ edge[50,51]. For in situ oxidation and temperature-dependent investigations, Si wafers with native oxide layer ($SiO_x$) are used as substrates, Fig. 1b. Samples suitable for ex situ HAADF-STEM studies are prepared under similar conditions on electron transparent $Si_3N_4$-membranes with a thickness of 10 nm covered with an amorphous carbon (a-C) layer to improve the electric conductivity required in XPEEM, as illustrated in Fig. 1c. As determined by the XA spectra, we found that we required approximately ten times higher oxygen exposure for the carbon-coated samples to achieve similar oxidation states as for the samples on the Si wafers. Upon controlled oxidation and XPEEM characterisation of the resulting magnetic and chemical state, the nanoparticles are covered with an additional amorphous carbon layer in UHV before exposing the sample to air for the transfer to the HAADF-STEM instrument, Fig. 1d. Further details about the sample preparation and the XPEEM data acquisition and analysis are provided in the "Methods" section.

Figure 1e shows an XPEEM elemental contrast map of a pristine, metallic Co nanoparticle sample on a Si/$SiO_x$ wafer. Bright spots originate from the resonant Co $L_3$ edge excitation of nanoparticles, which are randomly distributed across the substrate. The corresponding magnetic contrast map, shown in Fig. 1f, reveals that about half the nanoparticles are in a magnetically blocked (MB) state (see for instance the solid circles in the figure) and exhibit stable magnetic contrast ranging from black to white similar to earlier findings[40]. The magnetic contrast distribution in Fig. 1f reflects a random orientation of the magnetisation (**m**) of the nanoparticles due to the stochastic deposition process[40]. The other portion of nanoparticles is in a superparamagnetic (SPM) state (dashed circles), which is characterised by thermally induced fluctuations of **m** at a rate $\tau_r$ faster than the experimental acquisition time, $\tau = 400$ s in the present experiments. For $\tau \leq \tau_r$ ($\tau > \tau_r$), the nanoparticles are in a SPM (MB) state. We have shown earlier that the pristine Co nanoparticles in the MB state have a significantly enhanced magnetic anisotropy energy compared to bulk fcc Co[40]. Figure 1g shows that the in situ XA spectra extracted from both types of particles correspond to metallic Co and are practically identical, demonstrating that their different magnetic behaviour cannot be assigned to different chemical states (reference XA spectra are shown in Fig. S1 of the Supplementary Information). The metallic purity of the passivated nanoparticles in their pristine state is further confirmed by ex situ HAADF-STEM investigations as shown in Fig. 1h, demonstrating the absence of an oxide shell and confirming the effective passivation of the nanoparticles against air exposure by the a-C layers.

### Evolution of the chemical composition with oxygen exposure

The effect of oxygen exposure on the chemical state and magnetism of the nanoparticles at room temperature is shown in Fig. 2. Figure 2a–g

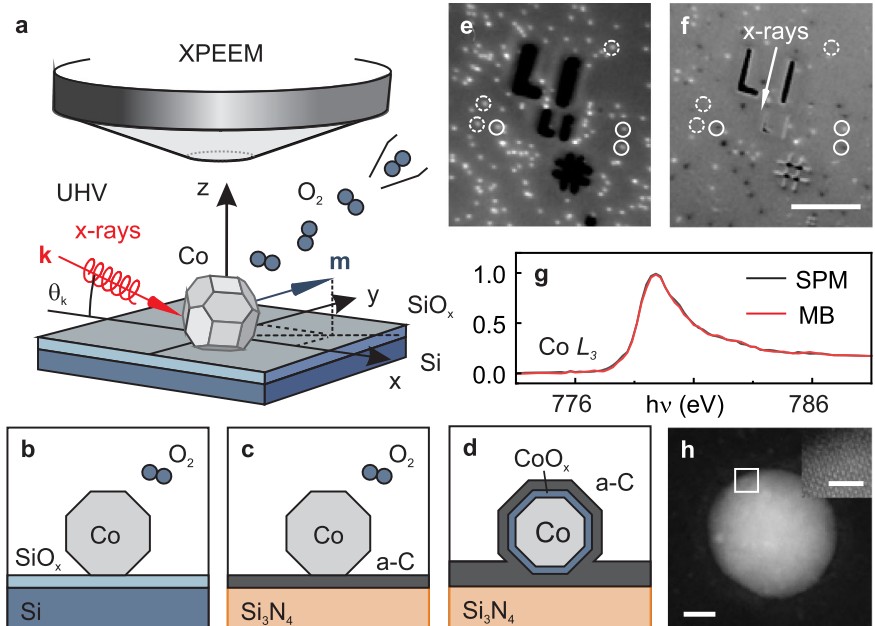

**Fig. 1 | In situ x-ray photoemission electron microscopy (XPEEM) investigation. a** Schematic of the XPEEM setup for the in situ oxidation of Co nanoparticles by means of molecular oxygen dosing under ultrahigh vacuum (UHV) conditions. **b**–**d** Illustration of the samples under investigation: **b** Co nanoparticles on Si/SiO$_x$ wafers, **c** on electron transparent Si$_3$N$_4$-membranes covered with an amorphous carbon (a-C) layer, and **d** upon capping with an additional a-C layer for transfer to the high-angle annular dark-field scanning transmission electron microscopy (HAADF-STEM) instrument. **e** XPEEM elemental and **f** respective x-ray magnetic circular dichroism (XMCD) contrast map of Co nanoparticles in the pristine metallic state on Si/SiO$_x$ acquired with the photon energy $h\nu$ set to the Co $L_3$ peak energy (the XMCD contrast ranges from ±0.025). The white arrow indicates the projected x-ray propagation direction. White circles indicate magnetically blocked (MB), dashed circles denote superparamagnetic (SPM) nanoparticles. The dark features are due to the saturated signal of lithographic gold marker structures used for sample navigation (scale bar is 3 µm). **g** Corresponding x-ray absorption spectra at the Co $L_3$ edge. **h** HAADF-STEM micrograph of a Co nanoparticle preserved in the pristine metallic state by means of the a-C layers (scale bar is 5 nm). The inset displays an enlarged image of the highlighted region (scale bar is 1 nm).

display the evolution of the XA spectra (black lines) up to an exposure of 20 L (1 L = 1.33 × 10$^{-6}$ mbar·s, see "Methods"). Figure 2a displays the XA spectra (average over SPM and MB) of the pristine state of the nanoparticles for better comparison. Exposure to about 1 L oxygen leads to the appearance of two noticeable shoulders on both sides of the metallic Co $L_3$ peak, Fig. 2c. These shoulders, here referred to as Co $L_{3a}$ and $L_{3c}$, correspond to XA features of CoO, see Fig. S1b. A third CoO peak, denoted as Co $L_{3b}$, coincides with the Co $L_3$ peak of metallic cobalt. Up to this oxidation stage, we find only minor changes in the magnetic contrast maps, see Fig. 2i–k. For instance, a few nanoparticles, such as particle "2", reverse their magnetic contrast due to thermally induced reversals of **m**[40,52]. Upon further oxygen exposure, the XA spectra evolve by the appearance of another peak at higher photon energy, as shown in Fig. 2d–g, which is referred to as Co $L_{3d}$ and indicates the formation of Co$_3$O$_4$, see Fig. S1c. These chemical changes are accompanied by a loss of magnetic contrast in most of the nanoparticles, for instance particles "1" and "2" in Fig. 2l–o. Only a few nanoparticles retain magnetic contrast at all exposures, see for instance particle "3" in Fig. 2i–o. Quantitative composition analysis of the XA spectra (described in the "Methods" section and in Fig. S2) reveals the proportion of Co, CoO, and Co$_3$O$_4$ as a function of oxygen exposure in the surface-near volume of the nanoparticles that is detected in XPEEM, which has a probing depth of 3–5 nm, see "Methods". In the composition analysis, we also included wurtzite-CoO (w-CoO), which can be stabilised in oxidised cobalt nanoparticles[21]. The data reveal an initial rapid increase of CoO, peaking at about 5 L before its relative proportion starts to decrease, see Fig. 2h. A small amount of w-CoO appears only at 10 L, possibly as a transition phase between CoO and Co$_3$O$_4$. The formation of Co$_3$O$_4$ occurs at a much lower rate when compared to CoO and increases monotonously, saturating at about 40 L. The contribution of metallic Co in the surface near region drops rapidly till 5 L and approaches zero at the highest

dosages. Based on this evolution and the respective signature in the magnetic contrast maps, we identify three different chemical states as follows: (A) pristine metallic state, (B) CoO-dominated state, and (C) saturated, Co$_3$O$_4$-dominated state, whose onsets are labelled in Fig. 2.

## Local atomic structure
Representative HAADF-STEM micrographs of different Co nanoparticles in states A–C embedded in a-C on Si$_3$N$_4$ membranes are shown in Fig. 3. The chemical and magnetic characterisation of the samples using XPEEM is shown in Fig. S3. The solid and open white circles represent, respectively, particles with and without XPEEM magnetic contrast in the state before carbon capping. The left column displays Co nanoparticles in the pristine metallic state A. The middle column of Fig. 3 displays Co nanoparticles in oxidation state B that exhibited magnetic contrast (MB state) before oxidation. As indicated, the bottom two nanoparticles lost their magnetic contrast upon oxidation. The HAADF-STEM data yield no obvious morphological or structural characteristics that would explain the different magnetic behaviour. Instead, in all cases we find a defect-rich oxide shell with a thickness of about 4 nm around the remaining metallic core. Some larger defects in the oxide shell are highlighted with white arrows. The right column of Fig. 3 shows Co nanoparticles in state C where magnetic contrast is lost in nearly all nanoparticles (see Fig. 2). Most obvious in this state are the extended voids between the metallic core and the oxide shell, indicating the onset of the nanoscale Kirkendall effect[10]. Furthermore, the metallic core size is reduced, while the oxide shell thickness is approximately the same as in state B. Like in state B, we do not find a correlation between structural features and the magnetic state before and after oxidation.

A closer inspection of the HAADF-STEM data reveals further structural details in the nanoparticles and their oxide shells. Specifically, we find that most of the metallic nanoparticles in state A possess

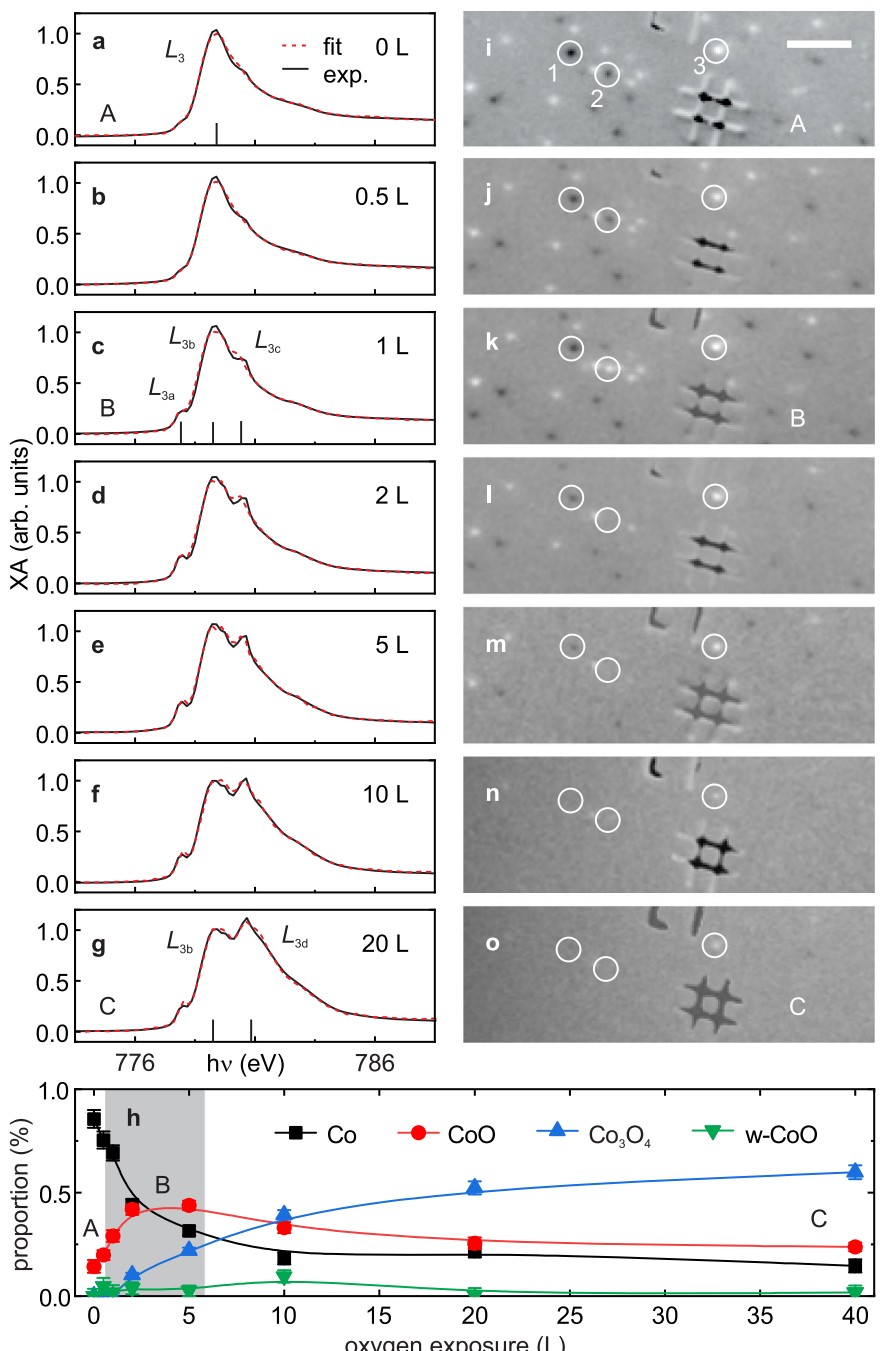

**Fig. 2 | Evolution of the chemical and magnetic states of Co nanoparticles deposited on Si/SiOₓ during in situ oxidation. a–g** X-ray absorption (XA) spectra with increasing molecular oxygen dosage in langmuir (L) (black lines). Specific features at the Co $L_3$ edge of metallic Co and of the different cobalt oxides discussed in the text are indicated in **a**, **c**, **g**. Red dashed lines are fits to the XA spectra. **h** Relative proportion of Co, CoO, wurtzite-CoO (w-CoO), and $Co_3O_4$ as a function of oxygen dosage obtained from a chemical composition analysis. The lines are guides to the eye. **i–o** X-ray magnetic circular dichroism contrast maps for the different oxygen dosages. The scale bar is 500 nm. The three specific chemical states, A–pristine, metallic state, B–CoO-dominated state, and C–$Co_3O_4$-dominated state are denoted in the respective panels. A movie sequence of **i–o** showing the evolution of the magnetic contrast as a function of oxygen dosage over a larger sample area of 20 μm field of view is available as Supplementary Movie 1.

face-centred-cubic (fcc) structure and not the hexagonally closed-packed lattice known from bulk Co at room temperature. Moreover, most of the nanoparticles are not single crystalline, but consist of several grains or possess structural defects, as frequently reported for fcc Co nanoparticles[40]. One example is given in Fig. 4a, where regions "i" and "ii" correspond to fcc Co grains with different crystallographic orientations. As exemplified by the enlarged, atomic level resolution of a nanoparticle in state B shown in Fig. 4b, we find that nanoparticles in state B possess oxide shells that consist of many small crystallites with

typical sizes of 3–5 nm. The atomic resolution HAADF-STEM data allow us to probe possible crystallographic relations between the metallic Co cores, the CoO, and the $Co_3O_4$ oxide crystallites. In particular, we find no specific orientation of the CoO crystallites relative to the lattice of the metallic Co core grains, suggesting that the nucleation of oxide crystallites occurs with a random orientation relative to the metallic Co core. The oxide shells of nanoparticles in state C reveal as well CoO and $Co_3O_4$ crystallites, where CoO crystallites are also found in the outer region of the shell.

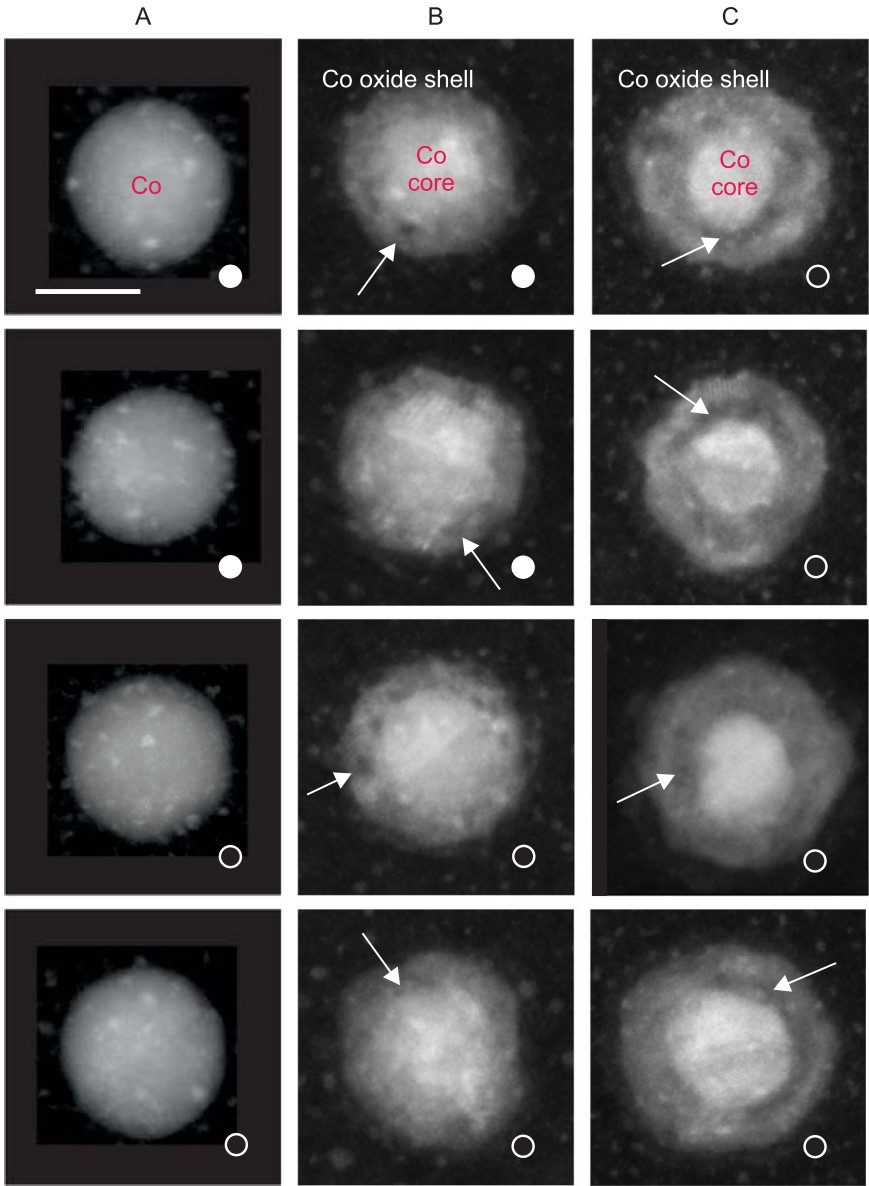

**Fig. 3 | High-angle annular dark-field scanning transmission electron microscopy (HAADF-STEM) images of carbon-capped Co nanoparticles in states A–C on Si₃N₄ membranes (corresponding respectively to 0 L, 1 L, and 20 L on Si/SiOₓ of Fig. 2).** The micrographs in each column correspond to the same oxidation state as indicated by the labels. The white arrows highlight specific defects. Solid (open) white circles indicate that the respective nanoparticle exhibits (no) magnetic contrast in x-ray photoemission electron microscopy (XPEEM) in the indicated chemical state. The scale bar is 10 nm.

## Magnetic state at lower temperatures

To further study the correlation between the magnetic properties and composition of the oxide shell, we perform temperature-dependent in situ XPEEM experiments on nanoparticles in different oxidation states. Figure 5a shows the XA spectra (black line) of pristine nanoparticles (state A). The corresponding XPEEM image and the magnetic contrast map are shown in Fig. 5d, g, respectively. As already shown above (Fig. 2), oxidation to state B, Fig. 5b, results in a reduction of magnetic contrast in a large number of nanoparticles, Fig. 5e. Cooling the sample to 114 K leads to a recovery of magnetic contrast in most of the nanoparticles. The vast majority of the nanoparticles exhibits the same magnetic contrast (black or white) upon oxidation and cooling as in the pristine, metallic state, e.g., particles "3" and "4" in Fig. 5h, g. Increasing the amount of Co₃O₄ by further oxidation to state C results in a further loss of magnetic contrast at room temperature, Fig. 5f, and almost no recovery of contrast at 114 K, Fig. 5i, and down to 58 K (shown in Fig. S4).

## Discussion

The present investigation demonstrates that pure Co nanoparticles oxidise in a two-step process with immediate CoO formation and subsequent conversion to Co₃O₄, very similar to what is found for Co thin films or in bulk[27]. Our data show that these oxidation steps occur at room temperature, at very low O₂ partial pressure, and at very low oxygen doses, comparable to the oxidation of ultrathin Co films under ultrahigh vacuum conditions and as expected given the reactivity of 3d transition metals and the high number of low-coordinated surface atoms. Based on this finding, we attribute the much lower reactivity of Co nanoparticles towards oxidation found in earlier reports[2,18,21,24,26,35–37], requiring often considerably elevated temperatures and (near) atmospheric oxygen pressure for the transformation of metallic cobalt into an oxide phase, to the partial or full passivation of the nanoparticle surface with uncontrolled pre-oxidation or chemical agents such as oleic acid used as capping ligands which block the access of molecular oxygen to the surface of the nanoparticle. Hence,

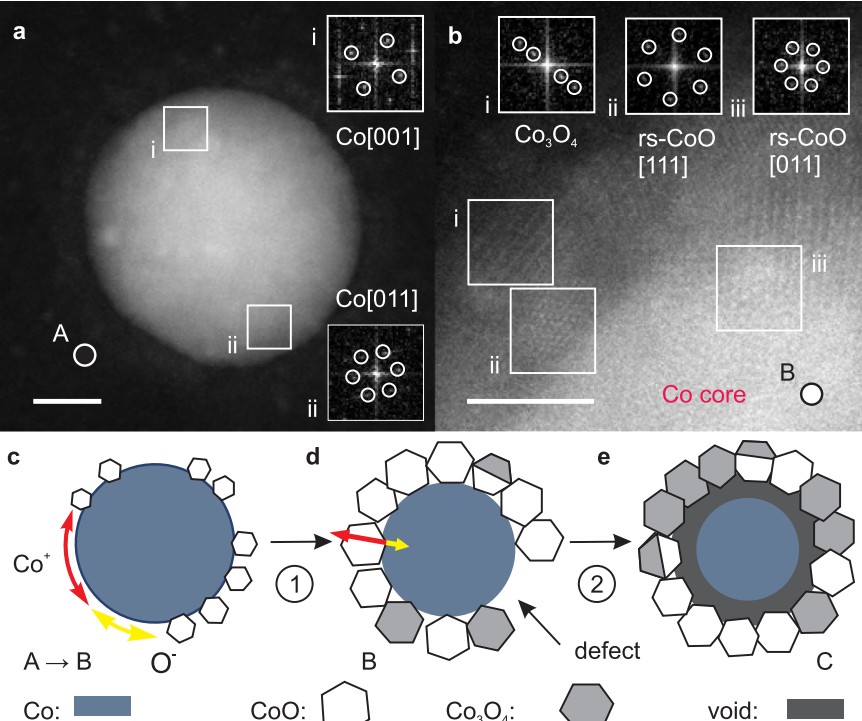

**Fig. 4 | Atomic structure and atomistic oxidation model. a** High-angle annular dark-field scanning transmission electron microscopy (HAADF-STEM) image of a pure metallic nanoparticle (state A); insets (i) and (ii) show fast Fourier transforms (FFTs) and the deduced zone axes of the areas with the same labels marked by white squares (scale bar is 5 nm). **b** Enlarged image with atomic-level resolution of a nanoparticle in state B; the insets (i–iii) show FFTs of the areas with the same labels marked by white squares revealing one $Co_3O_4$, (i), and two CoO crystallites, (ii) and (iii), with different orientations (scale bar is 2 nm). Notice the preferred orientation between CoO and $Co_3O_4$ crystallites in region "i" and "ii", where the {110}-planes of CoO and {110}-planes of $Co_3O_4$ are parallel, but without a specific in-plane orientation of the crystal axes. **c–e** Model of the oxidation process showing the onset of the different oxidation states, A, B, and C.

to exploit the full chemical reactivity of pure metallic Co nanoparticles, the preparation of bare, unpassivated and/or ligand-free surfaces is critical. For example, thermal decomposition is employed often to remove undesired oxide shells from the particle surface, motivated by the thermal instability of the 3d metal oxides. However, earlier in situ experiments revealed that such an approach for Co nanoparticles, even under additional reductive exposure to near ambient pressure of molecular hydrogen, does not fully remove the oxide shell, which was ascribed to the possible formation of a stable w-CoO nanophase[21]. A more promising approach was demonstrated based on exposure to a hydrogen plasma, which can be preceded by an oxygen plasma exposure if organic shells need to be removed[24,25]. In order to avoid here any ambiguities in the chemical nature of the particles entering interrogations with oxygen, metallic ligand-free cobalt nanoparticles were deposited under UHV conditions, free of contaminants, and directly entered into the oxidation experiments.

We may further compare the thickness of the observed oxide shells with the thickness of comparable oxide layers in thin films[27]. Upon exposure to a few langmuir of molecular oxygen, thin films form typically oxide layers with a thickness in the order of 1–2 nm[53]. In contrast, the present Co nanoparticles form a 4-nm thick CoO shell upon exposure to 1 L molecular oxygen on $Si/SiO_x$, see Figs. 2 and 3. Such thick oxide shell in a comparable chemical state may indicate a higher reactivity when compared to thin films due to the larger surface-to-volume ratio. However, assuming that each oxygen molecule hitting the nanoparticle surface is chemisorbed and contributes to the formation of the oxide shell, one finds that a nominal exposure of 6 L molecular oxygen is required in order to form a 4 nm CoO dominated shell (see Section 1 of the Supplementary Discussion). This strongly indicates that the substrate plays a major role in the oxidation of supported nanoparticles via surface diffusion of adsorbed oxygen,

similar to spillover effects known from other chemical surface reactions[54]. For $SiO_x$, we estimate an oxygen diffusion length of 200 μm at room temperature, such that diffusion of oxygen physisorbed on the substrate over an area nearly $2 \times 10^9$ larger than the area of the nanoparticle can be directly adsorbed on the surface of the Co nanoparticles as discussed in Section 1 of the Supplementary Discussion. Further evidence for the role of the substrate comes from the observation that we need approximately ten times higher oxygen exposures to achieve oxidation states B or C on the carbon-coated $Si_3N_4$ membranes when compared to the Si wafers. Additional evidence can be adduced from previous in situ oxidation studies under ultrahigh vacuum conditions. For instance, in well-separated Co nanoparticles deposited onto a clean, but chemically reactive Ni film, an oxygen dose approximately two orders of magnitude higher than in the present work was required to reach an oxidation of the Co nanoparticles similar to the present state C[38]. In another example, in situ oxidation of a dense array of nanoparticles on a chemically inert Si-substrate[24] required a dosage of molecular oxygen for oxidation three orders of magnitude higher compared to the present work, while ex situ experiments showed that a much higher amount of oxygen is required for the oxidation of dense nanoparticle arrays than for isolated nanoparticles[19]. The latter findings can be explained by the fact that most of the substrate surface is covered with nanoparticles, such that oxygen diffusion through the substrate is strongly hindered.

In most previous investigations, a uniform, closed oxide shell with a thickness that grows with oxygen exposure is assumed and discussed in terms of the metal oxidation theory of Cabrera and Mott[49]. However, our data show that in the early stages of oxidation (state B) the oxide shell is not uniform. Instead, the oxide shell consists of randomly oriented CoO grains with sizes matching well the thickness of the oxide shell. In addition, as seen in the middle column of Fig. 3, the shells can

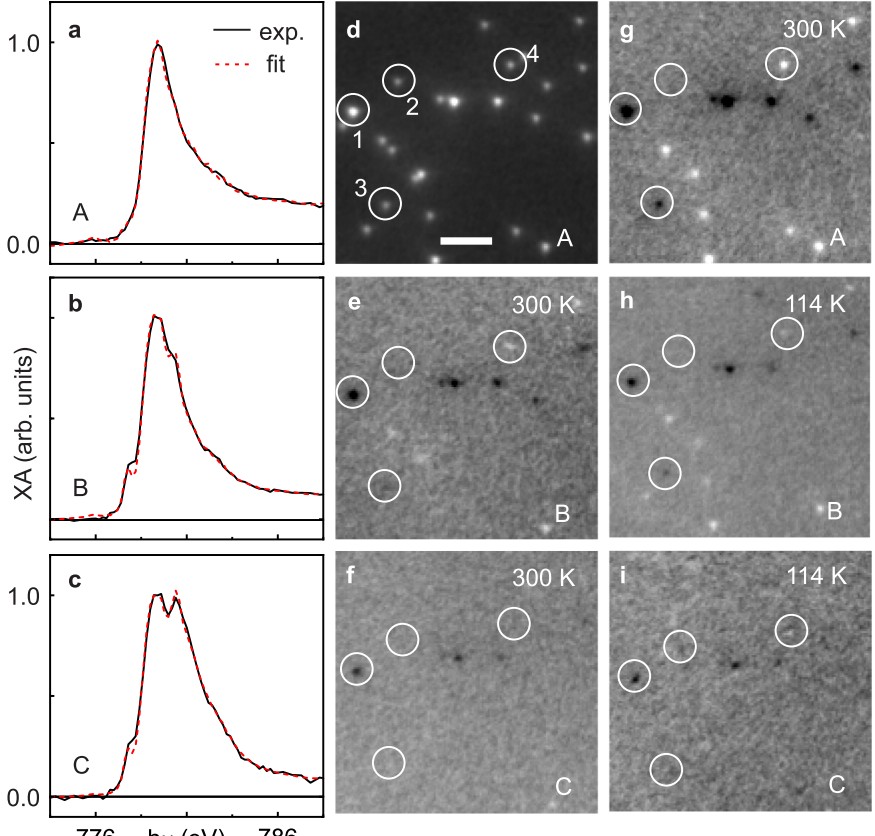

**Fig. 5 | Temperature dependent magnetic state of pure and oxidised nanoparticles. a–c** X-ray absorption (XA) spectra of Co nanoparticles at states A, B, and C (black lines; dashed red lines are fits to the data). **d, g** X-ray photoemission electron microscopy (XPEEM) Co elemental map and x-ray magnetic circular dichroism (XMCD) contrast map of the nanoparticles at 300 K in state A. **e, h** XMCD contrast maps of the nanoparticles upon oxidation to state B at 300 K and at 114 K and **f, i** in state C at 300 K and at 114 K. The scale bar in **d** is 1 μm. The XMCD contrast in **e–i** ranges from ±0.020. The XA spectral analysis yields a composition of Co (92 ± 5%), CoO (5 ± 3%), w-CoO (3 ± 3%), and $Co_3O_4$ (0 ± 2%) for the sample in state A, Co (50 ± 3%), CoO (44 ± 3%), w-CoO (1 ± 3%), and $Co_3O_4$ (5 ± 2%) in state B, and Co (27 ± 3%), CoO (34 ± 2%), w-CoO (1 ± 3%), and $Co_3O_4$ (38 ± 2%) in state C.

exhibit large defects. Moreover, the oxide shell thickness is practically the same in states B and C, although significant oxygen uptake and chemical reactions take place. These findings strongly suggest that the initial oxide shell formation occurs through independent nucleation and growth of CoO crystallites, rather through a uniform and closed oxide shell with a thickness that increases with the oxygen dosage. The extended defects also indicate that the probability for the formation/nucleation of oxide crystallites is not uniform at the nanoparticle surface, as expected from the varying surface atomic coordination and additional possible surface modifications induced by structural defects in the pristine metallic nanoparticles. Furthermore, the sizeable lattice mismatch between fcc Co and CoO is expected to prevent a uniform and/or epitaxial oxide shell growth and might promote the growth of small crystallites. Such a growth mode explains also the immediate formation of CoO in the first oxidation phase, Fig. 2h, since oxygen and metal ions can freely diffuse over the surface of both the metallic nanoparticle and the growing oxide crystallites, as illustrated in Fig. 4c. This efficient material transport stops when the growing CoO crystals merge to form a closed CoO shell as in state B. At this stage, the metallic core is largely passivated, and subsequent chemical reactions occur mostly as conversion of CoO to $Co_3O_4$ via ion diffusion through the oxide shell, which is much slower, Fig. 4d. The onset of the nanoscale Kirkendall effect in state C confirms that the conversion to $Co_3O_4$ is dominated by Co ion diffusion through the oxide layer, which might be driven by the electric field-induced ion diffusion mechanism through the oxide shell as proposed for the low temperature oxidation of thin films by Cabrera and Mott[10,18,49]. The random orientation of

oxide crystals observed in this work is not compatible with the formation of a uniform fcc oxygen lattice, in which the local Co ion concentration determines the nature of the oxide, CoO or $Co_3O_4$, respectively, as speculated earlier[21]. Our data show that the conversion is a process linked to the individual oxide grains, perhaps dependent on their orientation and the resulting surface facets exposed, see Fig. 4b, rather than a gradual, uniform process across the entire oxide shell surface. As a result, the oxide shell possesses a complex structure with phase pure CoO or $Co_3O_4$ and mixed phase oxide crystallites as illustrated in Fig. 4e. The same oxidation behaviour is observed in nanoparticles with sizes ranging up to about 30 nm as demonstrated in Fig. S5, highlighting that the present findings are also applicable for larger nanoparticles. (A discussion of the anticipated oxidation behaviour of nanoparticles smaller than 10 nm as deduced from the proposed two-step oxidation model and a comparison with reports in the literature is provided in Section 2 of the Supplementary Discussion.) We also note that exposure to a higher oxygen pressure is expected to lead to an increased number of nucleation sites and therefore, to a larger number of growing CoO crystallites which will coalesce at a smaller size, resulting in a somewhat thinner oxide shell. Indeed, previous work shows that similar Co nanoparticles exhibit a thinner oxide shell upon ambient air exposure[40]. Another important aspect of the oxidation mechanism concerns its temperature dependence. Increasing the temperature might increase not only the oxidation rate, provided that a sufficient amount of reactants is available but also assist in overcoming additional reaction barriers. Indeed, the literature shows that temperatures higher than about 550 K promote the formation of

$Co_3O_4$ in oxidised Co nanoparticles[21,23,37]. In addition, the effect of temperature on oxygen transport via the substrate should be taken into account. For chemically inert substrates, the latter is given by a competition between adsorption, desorption, and diffusion. For instance, the oxygen desorption temperature for Si wafers with a native oxide layer is about 440 K, while for carbon it ranges from 47 K for physisorbed molecular oxygen to higher temperatures for atomic oxygen, as highlighted in Section 1 of the Supplementary Discussion. The evolution of the oxidation state of cobalt, as well as of the morphology of the particles can have pronounced effects in catalytic reactions, up to defining both catalyst activity and selectivity in reactions that are structure sensitive and/or where the oxidation state of cobalt dictates performance, like in Fischer-Tropsch synthesis[55], dehydrogenation reactions[56] or combustion of hydrocarbons[57].

As shown in Fig. 5, oxidation to state B leads to a loss of magnetic contrast in a large portion of the nanoparticles at room temperature. Since the oxide shells of the individual nanoparticles develop with similar morphology and thickness and a number of nanoparticles in this chemical state still exhibit magnetic contrast, we conclude that the loss of magnetic contrast is not due to screening of the signal of the ferromagnetic metal core by the growing oxide shell and the limited XPEEM probing depth, see Methods. Further evidence for this can be concluded from the XA spectra, which reveal that in state B, about 30% of the signal originates from the metallic core, see Fig. 2h. Hence, the loss of magnetic contrast could be either due to the onset of superparamagnetic fluctuations or due to the occurrence of disordered spins in the probed volume of the nanoparticles. The experiments show that the magnetic contrast nearly recovers when cooling the samples to 114 K with the sign (i.e. dark or bright) being the same at low temperature as in the initial (metallic) state. Hence, the data exclude the onset of superparamagnetic fluctuations due to oxidation, which would lead to a random magnetic contrast at low temperature, and indicate instead a magnetically blocked ferromagnetic core. Support for this interpretation comes from earlier ensemble measurements, which suggested that oxidation leads even to the enhancement of magnetic energy barriers in the ferromagnetic cores of Co nanoparticles[2,25,35]. This scenario would further imply that the main contribution to the magnetic energy barrier of the metallic Co nanoparticles resides in the inner core of the particle and that the surface does not dominate the magnetism of nanoparticles, as frequently assumed in the literature[40].

The loss of magnetic contrast at room temperature could be assigned to the onset of oxygen-induced magnetic disorder at the oxide-shell-metal-core interface as postulated in earlier works[2,35]. Such disorder could be related to an interfacial reacted layer at the metallic Co core and a related perturbation of the local exchange interaction. The recovery of magnetic contrast in our data shows that such disorder could be partially overcome at lower temperatures. Taking advantage of the x-ray magnetic linear dichroism (XMLD) effect at the Co $L_3$ edge we have further addressed possible antiferromagnetic order in the oxide shell (see "Methods" and Fig. S6). The data reveal no sign of antiferromagnetic order in the oxide shell at 114 K although the small CoO crystallites in the oxide shell of oxidised Co nanoparticles have been found to order below $T = 235$ K, which is only slightly below the Néel temperature of bulk CoO[36]. We assign the absence of an XMLD effect at 114 K either to a lack of uniform alignment of the antiferromagnetic spin systems as suggested by the randomly oriented CoO grains across the shell or to a possible SPM state of the CoO oxide crystals due to their small size. Further oxidation to state C eventually results in a loss of magnetic contrast in nearly all nanoparticles at room temperature. At 114 K a few nanoparticles still exhibit weak magnetic contrast with the same sign as before oxidation, see circles in Fig. 5. No further changes are observed down to 58 K (Fig. S4). The weaker magnetic contrast when compared to state B at low temperature can be assigned to the reduced volume of the ferromagnetic core and the

respectively smaller metallic Co signal when compared to state B (Fig. 3). Also the physical separation of the metallic core and the oxide shell due to the Kirkendall effect might play a role in the attenuation of the metallic core signal.

Finally, we argue that the complex evolution of the structure of the oxide shell needs to be fully taken into account when discussing the magnetic properties of oxidised Co nanoparticles. In particular, we propose that in state B the random orientation of the CoO crystals and the absence of a uniform antiferromagnetic spin order in the oxide shell, is responsible for the reported reduced exchange bias of about 1 T in comparison to the theoretically expected 5–6 T. In state C, the physical and magnetic decoupling between the ferromagnetic core and the oxide shell due to the Kirkendall effect may explain the reduction of the exchange bias fields at higher oxidation states[2]. Further, a superparamagnetic contribution to the magnetisation curves of oxidised Co nanoparticles was found that was assigned to the presence of small metallic Co clusters in the interface region between the metallic core and the oxide shell[2]. Such clusters could be related either with metallic Co filaments that connect the metallic core with the oxide shell upon the onset of the Kirkendall effect and/or with CoO crystals that are converted partially to $Co_3O_4$[10]. Several reports have predicted a ferromagnetic layer at the interface between CoO and $Co_3O_4$[58,59]. Our data show that, in state C, mixed phase $CoO$-$Co_3O_4$-crystals exist and therefore, such ferromagnetic interface layers could be present. Due to their expected small size they would be likely in a superparamagnetic state.

In conclusion, we have demonstrated that pure Co nanoparticles have a chemical reactivity to molecular oxygen which is comparable to that of clean in situ prepared films and bulk surfaces at room temperature under ultrahigh vacuum conditions. Hence, their actual reactivity is orders of magnitude higher when compared to previous studies on Co nanoparticles and further enhanced via oxygen diffusion over the substrate surface. We identify a two-step oxidation mechanism whereby a rapid CoO shell formation is followed by a slower conversion to $Co_3O_4$, i.e., longer oxygen exposures results in only small increases in the oxide shell thickness. The lattice mismatch between metallic Co and CoO prevents the formation of a homogeneous oxide shell. Instead, CoO crystals nucleate and grow independently until they merge and form a closed layer. The large surface area emerging in the first growth mode promotes oxygen adsorption and Co ion diffusion and results in the very rapid formation of the CoO shell. In the second phase, outwards Co ion diffusion dominates the oxidation kinetics, while inward oxygen diffusion might lead to the formation of a layer with disordered spins in the metallic core. This layer and the random orientation of the CoO crystals relative to each other and to the metal core is likely responsible for the limited exchange bias observed in other reports. Further oxidation results in the nanoscale Kirkendall effect, which effectively decouples the oxide shell from the metallic core, explaining the reported reduction of the exchange bias effect upon advanced oxidation. Partial oxidation of the CoO crystals may lead to small superparamagnetic inclusions reported in the literature. Finally, our data demonstrate the absence of a pressure gap in the oxidation of Co nanoparticles and provide a benchmark for an improved understanding of the oxidation and magnetism of oxidised cobalt nanoparticles, with potential implications for their performance in catalytic reactions.

## Methods
### Sample preparation
Mass-filtered Co nanoparticles with sizes ranging from 8 to 20 nm are obtained from the gas phase by means of an UHV-compatible arc cluster ion source (ACIS) attached to a sample preparation chamber with a base pressure $<2 \times 10^{-10}$ mbar[43]. A gold mesh in the particle beam is used to control the particle density on the substrate, which is set to about one nanoparticle per μm$^2$ in order to achieve well-isolated

nanoparticles resolvable with the spatial resolution of about 50 nm of the XPEEM instrument and in order to prevent chemical and magnetic inter-particle interactions. The nanoparticles are deposited under soft landing conditions to avoid particle fragmentation or damage to the substrate[60]. Prior to the particle deposition, the substrates are annealed at 200 °C for 1–2 h in UHV to remove ambient air adsorbates from the surface. For the HAADF-STEM measurements, the Co particles are deposited onto transmission electron microscopy compatible $Si_3N_4$ membranes with a thickness of 10 nm, which are additionally covered with conductive amorphous carbon films with a thickness of 2–3 nm to facilitate XPEEM measurements on these substrates. Gold (platinum) markers are fabricated on the Si substrates ($Si_3N_4$ membranes) prior to the experiments using electron beam lithography (electron-beam-induced deposition in a focussed ion beam instrument) in order to identify the same particles for comparison between different measurement techniques.

### In situ XPEEM characterisation and data processing

After particle deposition, the sample is transferred under UHV from the deposition system to the XPEEM characterisation chamber with a base pressure of $2 \times 10^{-10}$ mbar. To oxidise the nanoparticles we introduce molecular oxygen (99.999% pure) into the XPEEM chamber through a leak valve, using partial pressures ranging from $2 \times 10^{-8}$ to $2 \times 10^{-6}$ mbar. The nominal dosages, $1 L = 10^{-6}$ Torr·s $\approx 1.33 \times 10^{-6}$ mbar·s, are estimated from the pressure reading in the XPEEM chamber. XA spectra and magnetic contrast images are acquired before and after each dosage to determine the change in chemical and magnetic states. During the oxidation reaction, the samples were not illuminated with x-rays. Consecutively recorded XA spectra and magnetic contrast maps showed no discernible x-ray induced chemical modifications to the nanoparticles. XPEEM characterisation is carried out at the Surface/Interface: Microscopy (SIM) beamline at the Swiss Light Source (SLS)[61]. XPEEM is a spectromicroscopy technique capable of providing spatially resolved spectroscopic data. In XPEEM, the local intensity of secondary electrons emitted from a surface near region (3–5 nm) of the sample excited by tunable, monochromatic x-ray light (proportional to the x-ray absorption) is measured to provide local maps of the XA of the sample with a lateral spatial resolution on the order of 50 nm[46]. The high voltage applied between the sample and the microscope objective lens used to accelerate the excited photoelectrons towards the imaging lenses is set to 15 kV for the samples using Si-substrates and to 10 kV for the $Si_3N_4$ samples in order to minimise eventual discharges that would lead to damage to the membrane. To obtain XA spectra, a sequence of images is taken at increasing photon energy, in the present case limited to the Co $L_3$ edge (770–790 eV); for better signal to noise ratio, the individual spectra of more than fifteen particles are averaged in our analysis. To identify the Co nanoparticles on the sample, we acquire separate images at the photon energy tuned to the $L_3$ edge of the metallic Co (779 eV) and a pre-edge energy (770 eV); pixelwise division of the two images give an elemental contrast image of the sample, where the Co nanoparticles appear as small dots with bright contrast in a grey background where no Co is present, see Fig. 1e. To obtain magnetic information, we use the XMCD effect corresponding to the difference in absorption for left and right circularly polarised light[50,51]. The magnetic contrast of an individual nanoparticle is given by the orientation of its magnetisation (**m**) relative to the x-ray propagation vector (**k**) according to **m·k**, see schematic in Fig. 1a. Magnetic contrast maps are obtained by pixelwise division of images acquired with right and left circularly polarised light ($I^-/I^+$) with the photon energy set to the metallic Co $L_3$ edge (779 eV) to give a direct measurement of the direction of the net magnetic moment on the sample; they are converted to XMCD maps by using the relation XMCD $= (1 - I^-/I^+)/(1 + I^-/I^+)$. The magnetic contrast maps are acquired using integration times of 10 s for each light polarisation; for better signal-to-noise ratio, 20 such images are acquired and averaged, resulting in a total experimental data acquisition time of $\tau = 400$ s. The magnetic relaxation time of the nanoparticles is given by an Arrhenius-type law $\tau_r = \tau_0 \exp(E_m/k_B T)$ with $1/\tau_0 = 1.9 \times 10^9$ s$^{-1}$ being the attempt frequency, $T$ the temperature, $k_B$ the Boltzmann constant, and $E_B$ is the magnetic energy barrier. The latter is determined by the magnetic anisotropy energy of the nanoparticles, which describes the preferred magnetisation axis and the energy required to reorient or reverse **m** within the nanoparticle. Possible antiferromagnetic order in the oxide shell is investigated by employing the XMLD effect at the Co $L_3$ edge.

### XA spectral analysis

XA spectra were analysed using spectral composition analysis[62] based on fitting a linear combination of reference spectra for metallic Co[63], CoO[64], w-CoO[21], and $Co_3O_4$ to the experimental data shown in Fig. S1[65]. The fitting equation is $Y(\text{total}) = P_1 Y(\text{Co}) + P_2 Y(\text{CoO}) + P_3 Y(\text{Co}_3\text{O}_4) + P_4 Y(\text{w-CoO}) + A + Bh\nu$ with $Y(X)$ being the reference spectrum of species $X$ with the respective weighting factor $P_i$. The term $A + Bh\nu$ is used to correct for small background contributions to the experimental data, where $h\nu$ is the photon energy. The Levenberg–Marquardt algorithm implemented in the non-linear least square fitting module in OriginPro is used to fit $Y(\text{total})$ to the experimental data by varying the parameters $P_i$, $A$ and $B$. From the fit results we calculate the relative proportions, e.g., $P(\text{Co}) = P_1/(P_1 + P_2 + P_3 + P_4)$. The error bars of $P(\text{Co})$, $P(\text{CoO})$, $P(\text{Co}_3\text{O}_4)$, and $P(\text{w-CoO})$ shown in Fig. 2h (typically smaller than the symbol size) are obtained by the propagation of the statistical errors of the $P_i$ obtained from the fits. As shown in Fig. S1, all four considered Co compounds exhibit very different XA spectra, which enable their identification.

### Structural characterisation

Atomic resolution HAADF-STEM characterisation was carried out using FEI Titan$^3$ microscopes, equipped with Cs probe correctors and operated at 300 kV, at EMAT at the University of Antwerp, Belgium. The beam current was set to 50 pA with a STEM pixel dwell time of 1–5 µs. Electron energy loss spectra (EELS) acquired in STEM revealed an oxide shell composition similar to the XA spectral analysis. The spectra were acquired at a dispersion of 0.25 eV per channel on a Gatan 977 spectrometer, with the monochromator of the microscope excited at 0.70. In addition, EELS maps were acquired using a beam current of 90–130 pA with an exposure time of 110 ms per pixel; to check for possible sample damage, two consecutive measurements on a number of particles were performed (with a smaller scanning step size for the second scan). These measurements revealed a sizeable reduction under the extended electron beam exposure in the EELS mode, see Fig. S7. No such effects were observed in HAADF-STEM imaging due to the much lower electron beam intensities and exposure times. These observations suggest that combining HAADF-STEM imaging with chemical analysis through in situ XA spectroscopy provides a non-destructive approach to investigating oxide shells of Co nanoparticles. Scanning electron microscopy was used to determine nanoparticle shapes and to distinguish individual nanoparticles from clusters of nanoparticles for the magnetic study on the Si/SiO$_x$ wafers. We note that besides spherical nanoparticles, we find also a small proportion of elongated nanoparticles typically composed of two or more merged spherical nanoparticles. While the shape plays an important role for the magnetic properties due to the shape anisotropy, we find no indication for a different chemical reactivity to oxygen (we also observe no size-dependent chemical behaviour in the investigated particle size range).

### Data availability

The raw data generated in this study have been deposited in the PSI Public Data Repository database under https://doi.psi.ch/detail/10.16907%2F054f5152-4ef9-4fa7-8a5b-3fb7ab838703. Source data are provided with this paper.

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

## Acknowledgements

This work is funded by Swiss National Foundation (SNF) (Grants. No 200021_160186 and 2002_153540) and the Swiss Nanoscience Institut (SNI) (Grant No. SNI P1502). S.V. acknowledges support from the European Union's Horizon 2020 research and innovation programme under grant agreement no. 810310, which corresponds to the J. Heyrovsky Chair project ("ERA Chair at J. Heyrovský Institute of Physical Chemistry AS CR - The institutional approach towards ERA"). The funders had no role in the preparation of the article. Part of this work was performed at the Surface/Interface: Microscopy (SIM) beamline of the Swiss Light Source (SLS), Paul Scherrer Institut, Villigen, Switzerland. We kindly acknowledge Anja Weber and Elisabeth Müller from PSI for their help in fabricating the sample markers. A.B. and J. Verbeeck received funding from the European Union's Horizon 2020 Research Infrastructure - Integrating Activities for Advanced Communities under grant agreement No. 823717 - ESTEEM3.

## Author contributions

A.K. and C.A.F.V. conceived the project. J. Vijayakumar, T.M.S., D.M.B., A.K., and C.A.F.V. performed the XPEEM experiments. J. Vijayakumar and T.M.S. analysed and interpreted the XPEEM data with support from A.K., C.A.F.V., and F.N. The HAADF-STEM measurements were performed by A.B., G.L., and J. Verbeeck. The HAADF-STEM data were analysed by A.K., J. Vijayakumar, G.L., and A.B. The manuscript was written by A.K., C.A.F.V., S.V., J. Vijayakumar, and T.M.S. with input from all authors.

## Competing interests

The authors declare no competing interests.
