## [Peer Review File · Nature Communications]

Absence of a pressure gap and atomistic mechanism of the oxidation of pure Co nanoparticlesREVIEWER COMMENTS

Reviewer #1 (Remarks to the Author):

Overview and general recommendation:

Studying seemingly simple oxidation reactions led to surprising and interesting results in the past especially when looking at nanoparticles of 3d transition metals. Within this area of interest, the nanoscale Kirkendall effect brought up many questions regarding mechanistic insights of oxidation processes. With the current manuscript the authors present interesting details of the initial room temperature oxidation mechanism of Co nanoparticles and the impact on their magnetism.

In general, the paper is well written and uses advanced experimental techniques such as in situ XPEEM, XAS, HAADF-STEM (conserved Co nanoparticles in different oxidation states) and cluster deposition. The results are new and presented in a clear and intriguing way. I have some general questions and minor comments which should be addressed before publication in Nature Communications:

2.1. Questions:

1. In the manuscript clusters in the size range between 10 and 20 nm are investigated. However, it is not clear to me how the size of the nanoparticles influences the observed oxidation mechanism. Furthermore, there is a minimum size for the observation of the Kirkendall effect, i.e. when particle size and crystallite size are in a similar range. How would the reported observations be applicable in this size range (below 10nm)? What happens with bigger nanoparticles?

2. The discussion of temperature effects such as higher mobility and diffusion length of atoms (important in the process of void formation) is missing. How would a higher and lower temperature influence the observations and conclusions?

2.2. Minor comments:

1. Line 134: MAE is not introduced before

2. Line 258: typo "than than"

3. Line 346: typo "the the"

4. Line 414: typo "nm nm"

5. Video: The video is (at least with my player) super quick (around 3s) and there is not enough time to follow all details (without stopping it several times). The usefulness of the video could be improved by marking 2 or 3 characteristic particles which the viewer should pay attention to. Otherwise, the knowledge to gain is rather limited by the video in the current state.

Reviewer #2 (Remarks to the Author):

The authors report on the oxidation of Co nanoparticles and their magnetic properties. This is an interesting fundamental work, providing valuable new data. The manuscript is well organized and written, and conclusions are for the most part well supported on experimental results. I recommend its publication after considering the following points:

1. Authors should discuss the effect of x-rays and the electron beam on the oxidation process

2. The authors should provide the fitting of the XA spectra shown in Figure 2. Where the small error in Figure 2h was calculated? How CoO was differentiated from wustite-CoO to obtain data shown in Figure 2h?

3. A somehow surprising conclusion of the work is that the void shell forms between CoO at the Co surface and the growing Co₃O₄ shell. What are the pieces of evidence supporting this conclusion? Data in Figure 4b is not conclusive taking into account the 3D structure of the shell.

Response to the Reviewers

We would like to thank the reviewers for their positive feedback and for the recommendation for our paper to be published in *Nature Communications* provided that a few aspects are addressed. In fact, we find the comments of the referees very helpful and we reply to them in detail below and in a revised manuscript (the text of the original reports below are set in blue/italics, our reply in black/normal type; references mentioned in this Reply correspond to those given in the manuscript). The manuscript has been revised to address the comments made by the reviewers (marked in blue) and we also took this opportunity to correct typos and improve the text readability. These revisions gave us the opportunity to further improve the presentation of our work, which we trust is now suitable for publication.

Reviewer #1 (Remarks to the Author):

Overview and general recommendation:

Studying seemingly simple oxidation reactions led to surprising and interesting results in the past especially when looking at nanoparticles of 3d transition metals. Within this area of interest, the nanoscale Kirkendall effect brought up many questions regarding mechanistic insights of oxidation processes. With the current manuscript the authors present interesting details of the initial room temperature oxidation mechanism of Co nanoparticles and the impact on their magnetism.

In general, the paper is well written and uses advanced experimental techniques such as in situ XPEEM, XAS, HAADF-STEM (conserved Co nanoparticles in different oxidation states) and cluster deposition. The results are new and presented in a clear and intriguing way.

I have some general questions and minor comments which should be addressed before publication in Nature Communications:

2.1. Questions:

1. In the manuscript clusters in the size range between 10 and 20 nm are investigated. However, it is not clear to me how the size of the nanoparticles influences the observed oxidation mechanism. Furthermore, there is a minimum size for the observation of the Kirkendall effect, i.e. when particle size and crystallite size are in a similar range. How would the reported observations be applicable in this size range (below 10nm)? What happens with bigger nanoparticles?

Response: We thank the reviewer for bringing this point, which was not addressed in detail in the original submission. In part, this was because of the constraints in the particle range that we are able to produce in our cluster source, but also because we saw no significant changes in behavior within the studied range, which in fact extended from 10-30 nm (although the number of such large particles

is low). For nanoparticles larger than those studied here, we do not expect a different behavior. Indeed, larger particles in our samples exhibit a very similar oxide shell evolution when compared to the smallest ones as demonstrated in Fig. R1, which compares representative HAADF-STEM image for smaller and larger nanoparticles in state B and state C.

Figure R1: (a) and (b) HAADF STEM images of oxidized nanoparticles in state B with different sizes as indicated. The image in (a) is the same as in the upper panel of the middle row in Fig. 3 of the main text. (c) and (d) HAADF STEM images of oxidized nanoparticles in state C. The scale bar is 5 nm.

For smaller nanoparticles, based on our findings, and in particular from the knowledge learned of the detailed oxidation process, we predict a size-dependent behavior as follows:

(i) As discussed in our manuscript, the first very oxidation step (CoO formation) from state A to state B is characterized by independent nucleation and growth of CoO crystals driven by very efficient surface diffusion of Co and oxygen until merging of the CoO crystals closes the shell. In the Supplementary Information of the original manuscript (see Section S1) we showed that a core-shell particle with a metallic core with a diameter of 11 nm surrounded by a 4 nm CoO shell requires an initial spherical metallic nanoparticle with a diameter of 16.4 nm. (For the sake of simplicity, we assume a homogeneous spherical shell.) This consideration suggests that the first oxidation step consumes a metallic layer thickness of about $(16.4 \text{ nm} - 11 \text{ nm})/2 \approx 3 \text{ nm}$. Assuming that the nucleation density for CoO crystallite growth and the required metal thickness does not depend sensitively on particle size, we can conclude that metallic Co nanoparticles with diameters up to $\approx 6 \text{ nm}$ are fully oxidized in the first oxidation step. This value corresponds indeed, as the reviewer points out, to the case where

the particle size (radius) and the oxide crystallite size are in a similar range and marks the lower particle size boundary for the onset of the Kirkendall effect.

(ii) The second, slower oxidation step towards state C is due to the Kirkendall effect, i.e., due to radial diffusion of oxygen and cobalt ions. To estimate how much Co is transferred from the metallic core to the oxide shell, we consider that the outer diameter of the void shell corresponds approximately to the original (in state B) size of the metallic core. (We emphasize that we do not have direct access with TEM to the very same particles in states B and C.) Our data (Fig. 3 of the manuscript) indicate that the thickness of the void shell is approximately 1 nm in width. Hence, we conclude that for nanoparticles with initial sizes between 6 and 8 nm the entire metallic core would be incorporated into the oxide shell after both oxidation steps to form a hollow oxide sphere, while for larger particles a yolk-shell structure results.

The predicted critical diameters discussed above (6 nm for fully oxidized nanoparticles, 6 – 8 nm for hollow oxide spheres, and yolk-shell or core-shell structures for nanoparticles larger than 8 nm) agree in fact well with findings reported in the literature, where nanoparticles oxidized under different conditions were investigated using TEM reported in Refs. 20, 23 and 37. We now include the above discussion in the revised manuscript in p.17, l.304-309 and added section S2 and Fig. R1 as Fig. S4 in the Supplementary Information.

2. The discussion of temperature effects such as higher mobility and diffusion length of atoms (important in the process of void formation) is missing. How would a higher and lower temperature influence the observations and conclusions?

Indeed, in the present work we did not address the temperature dependence of the oxidation mechanism: the detailed atomic processes might be very complex and include different diffusion activation barriers for different particles (including surface and bulk diffusion of oxygen, cobalt, and point defects) as well as different possible diffusion and reaction pathways, as discussed in Ref. 18, which we cannot directly address in our experiments. However, we can separate the temperature dependence in two regimes, a low temperature regime where no additional reactions occur and where the present results apply, and a high temperature regime where thermal activation opens new reaction and diffusion channels. For instance, for temperatures higher than about 550 K in Co nanoparticles conversion from CoO to Co₃O₄ is promoted and primarily Co₃O₄ is found, see e.g. Refs. 21, 23, and 37.

In addition, as discussed in our manuscript (p.15, l.251-270), we find that the substrate also plays a major role in the oxidation process, since oxygen adsorbed on the surface can diffuse to the nanoparticle and contribute to the oxidation, which is again a thermally activated process (we now

provide the diffusion constant length for oxygen on SiO₂ instead of Pt|Yttria of the original version, p.16, l.254-257). Concerning the dependence of the diffusion length with temperature, we start by considering the general expression for the diffusion length of a particle, given by $\delta = 2\sqrt{D\tau}$, where $D = D_0 \exp(-E_D/RT)$ is the diffusion coefficient and $\tau = \tau_0 \exp(E_\tau/RT)$ is the residence time until the particle is chemically bound or desorbed (where $1/D_0$, $1/\tau_0$ are attempt frequencies, E_D , E_τ are respectively the diffusion activation energy and the desorption or chemical reaction barrier, T is the temperature, and R the universal gas constant). Hence, $\delta = 2\sqrt{D_0\tau_0} \exp((E_D - E_\tau)/2RT)$ and its temperature dependence will depend strongly with the sign of $E_D - E_\tau$. In particular, for $E_D - E_\tau < 0$, corresponding to the case where the diffusion barrier is higher than the desorption energy, the diffusion length will be small and of the order of $2\sqrt{D_0\tau_0}$. In the opposite case, the diffusion length will be large at low temperatures and drop with increasing temperature, reflecting the fact that the adsorption energy is high, leading to a high residence time of the molecule of the surface, with the diffusion length strongly dependent on the diffusion barriers.

For the nanoparticle oxidation we can therefore identify three different atomic diffusion mechanisms that are likely to be strongly impacted by the temperature: (i) In the first oxidation step, where the surface diffusion of both oxygen and cobalt take place, the oxidation reaction rate is expected to be strongly affected. (ii) In the second oxidation step, after formation of the oxide layer, temperature-dependent differences in the bulk diffusion rates of oxygen and cobalt may influence the Kirkendall effect (as seen for instance in bismuth oxide hollow nanoparticle formation, Niu *et al.*, Nano Lett. 13, 5715 (2013)); in the extreme case where oxygen diffusion dominates, no Kirkendall effect would occur. (iii) The availability of oxygen reactant through the substrate will also be modified through a change in the diffusion length of the adsorbed oxygen that acts as source to the cobalt oxidation. We now discuss these aspects in the MS (p.18, l.313-322) and in the supplementary information, section S1.

2.2. Minor comments:

1. Line 134: MAE is not introduced before

2. Line 258: typo “than than”

3. Line 346: typo “the the”

4. Line 414: typo “nm nm”

5. Video: The video is (at least with my player) super quick (around 3s) and there is not enough time to follow all details (without stopping it several times). The usefulness of the video could be improved by

marking 2 or 3 characteristic particles which the viewer should pay attention to. Otherwise, the knowledge to gain is rather limited by the video in the current state.

We thank the reviewer for pointing out the typos mentioned above, which have now been corrected in the revised MS. With respect to the video, we have reduced its speed and marked 3 characteristic particles, as suggested.

Reviewer #2 (Remarks to the Author):

The authors report on the oxidation of Co nanoparticles and their magnetic properties. This is an interesting fundamental work, providing valuable new data. The manuscript is well organized and written, and conclusions are for the most part well supported on experimental results. I recommend its publication after considering the following points:

1. Authors should discuss the effect of x-rays and the electron beam on the oxidation process

Indeed, it is known that intense x-rays and electron beams can have a reducing effect and lead to exposure-induced changes in the x-ray absorption (XA) spectra (X-rays, XPEEM) or the EELS spectra (electrons, TEM) and, therefore, can have an impact on the oxidation process. To prevent such x-ray induced modifications during the oxidation process, the samples were only exposed to the x-rays when recording XA spectra or taking magnetic contrast images and not during oxygen dosing. In consecutively recorded datasets, we found no discernible x-ray induced changes. However, as described in the Method Section under “Structural characterization” and demonstrated in Fig. S3 (now Fig. S6) of the Supplementary Information of our original submission, we found a noticeable reduction effect in the EELS data acquisition at high electron beam intensities, which becomes particularly obvious in consecutively recorded EELS spectra, see Fig. S6 f and g. For the HAADF-STEM images, we used a much lower electron beam current and did not observe electron beam induced damages. We note that the oxidation reaction of these samples were carried out in XPEEM (with no x-rays on the sample) and the chemical reaction was therefore also not affected in STEM. These observations suggest that combining in situ XA spectroscopy with structural HAADF-STEM characterization presents the least destructive way to study the oxidation of Co nanoparticles. To clarify these aspects to the reader, we modified the text in p.22, l.434-437 in the revised manuscript.

2. The authors should provide the fitting of the XA spectra shown in Figure 2. Where the small error in Figure 2h was calculated? How CoO was differentiated from wustite-CoO to obtain data shown in Figure 2h?

Indeed, while in Fig.2 we provided the final result of the fit, we did not provide the individual components of the fit in order not to overload the figure; such spectral decomposition is presented in Fig.R2. We agree that such information may be useful to the reader and we now make it available in the Supplementary Information as Fig. S2 and refer to this figure in p. 9, l.158-162.

Figure R2 Composition of the XA spectra shown in Fig.~2 of the main text as deduced from the fitting procedure described in the Methods Section of the main text. Experimental XA spectra (circles), (1) linear background term ($A + B \text{ hv}$), (2) $Y(\text{w-CoO})$, (3) $Y(\text{Co}_3\text{O}_4)$, (4) $Y(\text{CoO})$, (5) $Y(\text{Co})$, and (6) $Y(\text{total})$.

We acknowledge as well that the details of the fitting procedure given in the methods sections were largely missing. The fitting equation is $Y(\text{total}) = P_1 Y(\text{Co}) + P_2 Y(\text{CoO}) + P_3 Y(\text{Co}_3\text{O}_4) + P_4 Y(\text{w-CoO}) + A + B \text{ hv}$ with $Y(X)$ being the reference spectrum of species X and P_i the respective weighting factor. The term $A + B \text{ hv}$ allows one to correct for small offsets in the experimental data that are linear in the photon energy hv . The Levenberg–Marquardt algorithm implemented in the non-linear least square fitting module in OriginPro is used to fit $Y(\text{total})$ to the experimental data by varying the parameters P_i ,

A, and B. From the fit results we calculate the relative proportions, e.g. $P(\text{Co}) = P_1/(P_1 + P_2 + P_3 + P_4)$. The error bars of $P(\text{Co})$, $P(\text{CoO})$, $P(\text{Co}_3\text{O}_4)$, and $P(\text{w-CoO})$ shown in Fig. 2h (and which are typically smaller than the symbol size) are obtained by the respective propagation of the statistical errors of P_i obtained from the fits. As shown in Fig. S1 of the Supplementary Information all four considered Co compounds exhibit very different XA spectra, which enable their identification. We have included this information to the manuscript in p.24, l.474-484, and in the caption to Fig.2. In particular, it can be observed in Fig. S1(b) that CoO shows a very pronounced multiplet structure which is absent in the spectrum of w-CoO shown in Fig. S1(d), which allows for a clear identification. This is now discussed in more detail in the revised manuscript (p.9, l. 158-162). We note that in our data the contribution of w-CoO is relatively small and appears possibly as a transition phase between CoO and Co_3O_4 , as discussed in the original manuscript (l.164-167).

3. A somehow surprising conclusion of the work is that the void shell forms between CoO at the Co surface and the growing Co_3O_4 shell. What are the pieces of evidence supporting this conclusion? Data in Figure 4b is not conclusive taking into account the 3D structure of the shell.

We thank the reviewer for drawing attention to this point, which may stem from the schematic displayed in Fig.4d-e in the original submission, showing a Co oxide layer between the metallic Co core and the void/oxide shell (the dark blue layer). In fact, this Co oxide layer was meant to represent only to an interfacial reacted layer at the metallic Co core (e.g. a chemisorbed oxygen layer) and not a well-defined oxide layer (hence our representing this layer as CoOx). With hindsight, we find that this representation may be misleading and we have omitted this layer from the model schematic shown in the revised Fig.4d-e.

To reiterate, our data indicate that the void layer forms between an outer $\text{CoO}/\text{Co}_3\text{O}_4$ shell and the metallic Co core. The evidence is provided in Fig.3, where we see the void regions separating the metallic Co core (displaying larger brightness) and the oxide shell. While the oxide shell of the particle presented in Fig.4b shows that the regions with Co_3O_4 lie at the external boundary and the CoO region within the oxide shell, this is found to be not always the case, with other particles also showing CoO crystallites in the outer region of the particle. In the region between the void and the Co metallic particle, one would assume some degree of surface oxidation, e.g., in form of a chemisorbed layer of oxygen, which is difficult to access deep in the particle volume in STEM. To make this point more clear and avoid any confusion, we have revised the manuscript both to omit the Co-O reaction exchange layer in Fig.4d-e and made the discussion about the oxide shell structure more clear by bringing the points mentioned above into the text in p. 13, l. 203-204 and p. 19, l. 350-351.

REVIEWERS' COMMENTS

Reviewer #1 (Remarks to the Author):

I appreciate the detailed answers of the authors to my questions as well the improvements to the manuscript. I am happy to recommend the article in the current state for publication!

Reviewer #2 (Remarks to the Author):

The authors properly took into account the reviewer's comments and corrected the manuscript accordingly. I recommend its publication in its current form

Response to the Reviewers

We would like to thank the reviewers for their positive feedback and for the recommendation for our paper to be published in *Nature Communications*.

Reviewer #1 (Remarks to the Author):

I appreciate the detailed answers of the authors to my questions as well the improvements to the manuscript. I am happy to recommend the article in the current state for publication!

Reviewer #2 (Remarks to the Author):

The authors properly took into account the reviewer's comments and corrected the manuscript accordingly. I recommend its publication in its current form.